# Genetic Correlation of Virulent *Salmonella* Serovars (Extended Spectrum β-Lactamases) Isolated from Broiler Chickens and Human: A Public Health Concern

**DOI:** 10.3390/pathogens11101196

**Published:** 2022-10-17

**Authors:** Ahmed Orabi, Wagih Armanious, Ismail A. Radwan, Zeinab M. S. A. Girh, Enas Hammad, Mohamed S. Diab, Ahmed R. Elbestawy

**Affiliations:** 1Department of Microbiology, Faculty of Veterinary Medicine, Cairo University, Giza 12211, Egypt; 2Department of Bacteriology, Mycology and Immunology, Faculty of Veterinary Medicine, Beni-Suef University, Beni-Suef 62511, Egypt; 3Department of Poultry Diseases, National Research Centre, Giza 12622, Egypt; 4Agricultural Research Center (ARC), Animal Health Research Institute-Mansoura Provincial Lab (AHRI-Mansoura), Giza 12618, Egypt; 5Department of Animal Hygiene and Zoonoses, Faculty of Veterinary Medicine, New Valley University, El Kharga 72511, Egypt; 6Department of Poultry and Fish Diseases, Faculty of Veterinary Medicine, Damanhour University, Damanhour 22511, Egypt

**Keywords:** *Salmonella* serovars, ESBL, resistance genes, virulence genes

## Abstract

This study aimed to detect the virulent *Salmonella* serovars (including ESBLs producing) isolated from broiler chickens and humans. Three hundred broilers and sixty human fecal samples were bacteriologically examined. Thirty (10%) and fourteen (23.4%) *Salmonella* isolates were recovered from broiler and human samples, respectively. The most predominant serovar was *S. enteritidis* and *S. typhimurium*. All *Salmonella* isolates were confirmed by conventional PCR-based *inv*A and *omp*A genes. Multidrug resistant (MDR) isolates were screened for the detection of *adr*A and *csg*D biofilm-associated genes, which were found in all isolated serovars except one *S. typhimurium* and 2 *S. infantis* of chicken isolates that were devoid of the *adr*A gene. Moreover, MDR isolates were screened for detection of seven resistance genes including ESBLs and other classes of resistance genes. Chicken isolates harbored *bla*_TEM_, *int*1, *bla*_CTX_ and *qnr*S genes as 100, 27.8, 11.1 and 11.1%, respectively, while all human isolates harbored *bla*_TEM_, *int*1 and *int*3 genes. The genetic correlations between virulent *Salmonella* serovars (including antimicrobial resistance) avian and human origins were compared. In conclusion, the high prevalence of virulent ESBL producing *Salmonella* serovars in broilers and humans with genetic correlations between them might be zoonotic and public health hazards.

## 1. Introduction

*Salmonellae* are very important bacterial pathogens affecting the poultry industry and human health [1]. Their principal habitat is the intestinal tract of humans, animals and birds, causing significant morbidity and mortality worldwide [2]. The Centers for Disease Control and Prevention (CDC) divided genus *Salmonella* into two species—*Salmonella enterica* and *Salmonella bongori* [3]. There are approximately 2600 serovars of *Salmonella* have been identified depending on the differences in their lipopolysaccharide layer with regard to their somatic (O) and flagellar (H) antigens [4]. The identification of *salmonellae* biochemically and serologically are time consuming; therefore, molecular methods based on DNA technology are recommended [5].

*Salmonellae* are invasive bacteria and their ability to infect and invade a host depends on the existence of many virulence factors. At least 60 genes are located on *Salmonella* pathogenicity islands (SPIs), which play the main role in *Salmonella* pathogenicity—especially SPI-1 and SPI-2—encoding structural proteins forming needle-like complexes allowing the insertion of the bacterial proteins into the host cells and modulating the cellular functions and immune pathways [1]. The invasion-related (*inv*A) gene is located within SPI-1 on the chromosome encoding inner membrane protein of the bacteria specific for invasion of the epithelial cells of the host [6]. The *inv*A gene is one of the most common PCR targeting genes and its amplification is considered an international standard for the determination of *Salmonella* [7]. Moreover, among outer membrane proteins (OMPs), *Omp*A is a major *Salmonella* antigenic protein, which is common to the majority of the stimulatory fractions; therefore, the *omp*A gene is highly specific for the detection of *Salmonella* serovars [8].

Antimicrobial misuse in therapy, prophylaxis, or as growth promoters has been implicated as a risk factor in the development and spreading of antimicrobial resistance (AMR) [1] leading to the limitation of therapeutic options available for the human salmonellosis treatment [9]. Β-lactams; penicillins and cephalosporins, are drugs of choice for the treatment of *Salmonella* [10]. The extended spectrum β-lactamases (ESBLs) and plasmid-mediated *amp*C β-lactamase are essential causes of antimicrobial resistance (AMR) [11]. In *Enterobacteriaceae*, resistance to cephalosporins is connected with the production of large spectrum β-lactamase as ESBLs and *amp*C β-lactamase [11]. ESBLs are widely reported all over the world and have been associated with successful enterobacterial clones having huge epidemic potential [12,13]. ESBLs are plasmid-mediated β-lactamase enzymes that hydrolyze penicillins, 3rd and 4th generation cephalosporins and monobactams with an oxyimino side chain [14]. ESBLs coding plasmids also carry resistance genes for other antimicrobial classes as aminoglycosides [15].

As it is very important to have knowledge about the behavior of *Salmonella* strains against antimicrobial agents, this study was planned to record the updates of virulence and ESBL among *Salmonella* serovars recovered from broiler chickens as well as of human origin and seeking the detection of correlations between broiler and human *Salmonellae*.

## 2. Materials and Methods

### 2.1. Ethical Approval

The study was approved from Beni-Suef University, Institutional Animal Care and Use Committee [BSU-IACU/ http://www.bsu.edu.eg (accessed on 1 January 2019)].

### 2.2. Samples

A total of 360 fecal samples (300 from broiler chickens and 60 from humans) were collected in plastic screw-top tubes containing sterile 10 mL buffered peptone water (BPW) (Oxoid Ltd., Basingstoke, UK) under complete hygienic conditions and transported in an ice box to the laboratory of the Faculty of Veterinary Medicine, Beni-Suef University, for bacteriological examination. Regarding the history of chicken samples, they were collected from 30 broiler chicken flocks resembling five Egyptian governorates during 2019–2020 with a total no. of 300,000 birds aged 1 to 14 days that were suffering from diarrhea, unabsorbed yolk sac, enteritis, arthritis and conjunctivitis. For human samples, all were collected from 60 patients aged 30–40 years during 2020, who were suffering from enteritis and hepatic problems.

### 2.3. Isolation, Identification and Serotyping of Salmonella Isolates

Next, 0.1 mL of BPW broth was transferred into tubes containing 10 mL Rappaport Vassiliadis broth (RVB) medium and was incubated at 41.5 °C for 24 h. A loopful from RVB was streaked onto xylose-lysine-deoxycholate (XLD) agar plates (Oxoid Ltd., Basingstoke, UK) and was incubated at 37 °C for 24 h. The colonies were examined for their cultural characteristics and morphological appearance. The suspected *Salmonella* colonies, red colonies with black centers, were subjected to Gram’s staining and biochemical identification [16]. Moreover, suspected *Salmonella* isolates were serotyped according to the Kauffman–White scheme using slide agglutination with standard “O” and “H” antisera (Difco, Sparks, MD, USA) [16].

### 2.4. Antimicrobial Susceptibility Testing

All *Salmonella* isolates were tested for their antimicrobial susceptibility against 13 different antimicrobial agents using the Kirby–Bauer disc diffusion method on Mueller Hinton Agar (Oxoid Ltd., Basingstoke, UK) according to the guidelines of CLSI [17]. The antimicrobials selected were those commonly used in the poultry industry, namely Cephalosporin group (Cefotaxime CTX, Ceftazidime CAZ, Ceftriaxone CRO, Cephalexin CL; 30 µg), Penicillins (Ampicillin AM; 10 μg, Ampicillin-Sulbactam SAM; 10 μg, Amoxicillin-clavulanic AMC; 30 µg), Carbapenems (Impenem IPM; 10 μg), Monobactams (Aztreonam ATM; 30 µg), Potentiated Sulfonamides (Sulfamethoxazole-trimethoprim SXT; 25 μg), Fluoroquinolones (Ciprofloxacin CIP; 5 μg) and Aminoglycosides (Amikacin AK; 30 μg, and Gentamicin CN; 10 μg) (Oxoid Ltd., Basingstoke, UK). Resistance to two or more antimicrobials of different classes was considered to be multidrug resistant (MDR) [1].

### 2.5. Conventional Polymerase Chain Reaction (cPCR) on Salmonella Isolates

All *Salmonella* isolates were confirmed by cPCR using *inv*A and *omp*A virulence genes. Moreover, MDR *Salmonella* serovars were screened for their existence in 2 *adr*A and *csg*D biofilm-associated genes as well as 7 resistance genes including ESBL genes; *bla*_TEM,_
*bla*_OXA-1_, *bla*_CTX_, *bla*_MOX_ and class 1, 2 and 3 integrons (*int*1, *int*2, *int*3), as well as the quinolone resistance gene (*qnr*A and *qnr*S).

Genomic DNA of the selected *Salmonella* isolates was extracted using the QIAamp DNA mini kit (Qiagen, Germany, Gmbh) according to the manufacturer’s instructions. The primers sequences and amplified products for the targeted genes for *Salmonella* isolates are depicted in Table 1.

## 3. Results

### 3.1. Occurrence of Salmonella Serovars in Chicken and Human Fecal Samples

Out of 300 broiler chicken fecal samples, 30 *Salmonella* isolates (10%) were recovered and serotyped. The most frequent serotype was *S. enteritidis* as six isolates (20%), then *S. infantis* (*n* = 5; 16.6%), each of *S. typhimurium*, *S. kentucky*, *S. heidelberg*, *S. hader* and *S. newport* (*n* = 3; 10% for each), *S. blegdam* (*n* = 2; 6.6%) and finally *S. maloe* and *S. geueletepee* (*n* = 1; 3.3% for each). Regarding the human fecal samples (*n* = 60); 14 *Salmonella* serotypes (23.4%) were identified and serotyped as five *S. enteritidis* and *S. typhimurium* (35.7% for each), three *S. anatum* (21.4%) and one *S. derby* (7.1%) (Table 2).

### 3.2. Antimicrobial Susceptibility Profiles

The results in Table 3 show that broiler chicken *Salmonella* serovars exhibited resistance to imipenem (83.3%), cefotaxime (80%), ceftriaxone (73.3%), aztreonam (70%), sulphamethoxazol-trimethoprim and ciprofloxacin (66.6% for each) and amikacin (60%). On the other hand, they showed sensitivity to cephalexin (73.3%) and gentamicin (70%). Eighteen *Salmonella* isolates (60%) were MDR. Meanwhile, human *Salmonella* serovars exhibited complete sensitivity to ceftazidime, amikacin and sulphamethoxazol-trimethoprim (100%) and high sensitivity to the other antimicrobials (78.5–92.8%). Four *Salmonella* isolates (28.6%) were MDR.

### 3.3. PCR for Screening Virulence and Resistance Genes in Salmonella Isolates

All the chicken and human *Salmonella* serovars were confirmed using PCR as they all harbored both *inv*A and *omp*A genes (100%). Regarding *adr*A and *csg*D biofilm-associated genes in MDR *Salmonella* serovars, all tested chicken isolates (*n* = 18) harbored the *csg*D gene (100%) while 15 (83.3%) harbored the *adr*A gene. Meanwhile, all human isolates harbored both genes (100%) (Table 4).

Regarding resistance genes in MDR *Salmonella* serovars, all tested chicken isolates harbored the *bla*_TEM_ gene (100%) while five (27.8%) harbored the *int*1 gene and two isolates harbored both *bla*_CTX_ and *qnr*S genes (11.1% for each). On the other hand, *bla*_OXA-1_, *bla*_MOX_, *int*2, *int*3 and *qnr*A genes were not found. Meanwhile, all human isolates harbored *bla*_TEM_, *int*1 and *int*3 genes (100%) while the other resistance genes were not found (Table 4).

### 3.4. Correlation between Virulence Genes and Antimicrobial Resistance among Salmonella Isolates from Broilers and Human

The results in Table 5 revealed that the virulence genes—*inv*A, *omp*A, *csg*D and *adr*A—existed in all MDR *Salmonella* serovars recovered from broiler chicken and human fecal samples except one *S. typhimurium* and two *S. infantis* isolates from broiler chickens, which did not have the *adr*A gene. A detailed analysis displayed an association of resistance phenotypes with potential virulence genes. The present study confirmed the relation between AMR and virulence genes in *Salmonella* serovars of avian and human origins.

## 4. Discussion

Avian Salmonellosis causes great morbidity, mortalities and consequently great economic losses in poultry industries with public health and zoonotic hazards [25]. This study included bacteriological examination of 300 broiler chicken and 60 human fecal samples and a molecular characterization of major virulence and AMR-associated genes. The obtained results showed that 30 *Salmonella* serovars (10%) were recovered from broilers while 14 serovars (23.4%) were recovered from 60 human fecal samples with a total percentage of 12.2%. The total prevalence was slightly lower than that reported by Tegegne [26] who recovered *Salmonella* from different sources with an overall prevalence of 16.9%. Regarding broilers, the obtained results were nearly similar to those of other studies [27]; 9.2%, and [28] 12%, while it was much lower than that recorded by other authors; [29] 30.5% and [26] 42.9%. The most frequent serovars in broilers were *S. enteritidis* (20%), *S. infantis* (16.6%), *S. typhimurium* and *S. kentucky* (10% for each). The most prevalent serotypes were *S. enteritidis*, *S. typhimurium*, *S. pullorum*, *S. vejle*, *S. amsterdam*, *S. infantis*, *S. petersburg*, and *S. atakpame* [30]. These results run parallel to those recorded by [1,28] as *S. enteritidis*, *S. typhimurium* and *S. kentuky* and *S. infantis* were the most frequent broiler chicken isolated serovars. For human samples, such prevalence was double that recorded in Ethiopia (11.3%) [26]. The most frequent human serovars were *S. enteritidis* and *S. typhimurium* (35.7% for each), *S. anatum* (21.4%) and *S. derby* (7.1%). These results coincided with those recorded by other authors [26,31] who recorded that *S. enteritidis*, *S. typhimurium* and *S. derby* were the most dominant non-typhoidal *Salmonella* serotypes in humans.

The virulence of *Salmonella* is related to its capability to invade a host cell, replicate and resist both digestion by macrophages and destruction by complement components of plasma [1,6]. The *inv*A gene-based PCR assay is considered an international standard procedure for the detection and confirmation of *Salmonella* spp. and mostly reduces the number of false-negative results commonly occurring in diagnostic laboratories [7]. Recently in Nigeria, 75% of *Salmonella* isolated from hospitalized febrile patients harbored the *inv*A gene [32]. Moreover, the *omp*A virulence gene plays an important role in bacterial adaptation to external environment stresses, which cause adhesion, invasion and host tissue damage [33]. The *omp*A gene was conserved among various *Salmonella* serovars and can be used, with high sensitivity and specificity, for Salmonella detection in food and clinical samples [8,34]. In our study, all the isolated *Salmonella* serovars were confirmed by cPCR using *inv*A and *omp*A virulence genes. Additionally, MDR *Salmonella* serovars were screened for the presence of *adr*A and *csg*D biofilm-associated genes which are responsible for cellulose production. The four tested genes were found in almost all chicken and human *Salmonella* isolates except for one *S. typhimurium* and two *S. infantis* chicken isolates which lack the *adr*A gene. The high expression of *adr*A and *csg*D genes, which are necessary for regulating cellulose production in *Salmonella* spp., showed a strong biofilm production capacity. These results are supported by those recorded by previous studies [35] in which all *Salmonella* isolates harbored the *inv*A gene while 90% contained *csg*D gene. Ćwiek et al. [36] recorded the prevalence of *adr*A and *csg*D as 100.0% for *S. enteritidis*. The obtained results were consistent with others who detected them in *S. enteritidis* and other *Salmonella* serovars [18,37,38]. The presence of these genes and their expression might be useful in *Salmonella* biofilm production via involvement in the synthesis of biofilm extracellular polymeric components [33]. Another study showed the presence of the *csg*D gene in all *Salmonella* isolates [39].

Antimicrobial agents are an effective tool for treating clinical diseases and maintaining healthy animals and human health. However, there is an increasing resistance of *Salmonella* to conventional therapeutic antimicrobial agents such as penicillins, sulfonamides, cephalosporins, amoxicillin-clavulanic and fluoroquinolones, which have been increasingly misused [31]. Antimicrobial-resistant *Salmonellae* create a public hazard, diminishing the therapeutic options available in the treatment of human salmonellosis [28]. In this study, 60% of broiler chicken *Salmonella* serovars were MDR, exhibiting resistance to β-lactams, cephalosporins, sulfonamides and fluoroquinolones. Meanwhile, only 28.6% of human *Salmonella* serovars were MDR, exhibiting lower resistance against β-lactams, cephalosporins, sulfonamides, fluoroquinolones and other antimicrobials used. These results resembled those recorded by other studies [31,36].

Antimicrobial resistance of *Salmonella* is one of the major virulence factors that enables the bacteria to survive and to be more pathogenic and toxigenic. ESBLs and plasmid-mediated *amp*C β-lactamase are essential causes of AMR [11]. ESBLs are one of the main causes of β-lactam resistance, including penicillin, 1st, 2nd and 3rd generations’ cephalosporins and monobactams by hydrolysis of antibiotics, among Gram-negative bacteria [40], and they can be inhibited by β-lactamase inhibitors such as clavulanic acid [14]. ESBLs coding plasmids may also carry additional β-lactamase genes and other resistance genes for other classes of antimicrobial [15]. This can restrict the treatment options for ESBL-producing pathogens and enhance the intra- and inter-species spreading of ESBLs [12]. There are many groups of ESBLs but those most frequently found in clinical Gram-negative isolates are *TEM*-, *SHV*-, *OXA*-, *CMY*- and *CTX-M*- types [1,41].

In the current study, all broiler chicken and human MDR *Salmonella* isolates were investigated using cPCR for seven resistance genes including ESBL and other AMR-associated genes (*bla*_TEM_, *bla*_OXA-1_, *bla*_CTX_, *bla*_MOX_, *int*1, *int*2, *int*3, *qnr*A and *qnr*S). Regarding broiler chicken isolates, only *bla*_TEM_, *int*1, *bla*_CTX_ and *qnr*S genes were represented as 100, 27.8, 11.1 and 11.1%, respectively. Meanwhile, all human isolates included *bla*_TEM_, *int*1 and *int*3 genes (100%). These results agreed with those obtained by Abdel Aziz et al. [42], who detected *bla*_TEM_ and *int*1 in all the tested *Salmonella* serovars recovered from broilers and humans. Additonally, the obtained results supported the fact that TEM-1 is the most popular plasmid mediated β-Lactamase in resistant Gram-negative bacteria [43]. The high *bla*_TEM_ gene frequency in our study also corresponded with Zhu et al. [44]. In other studies, the *bla*_TEM_ gene existed in 51.6% of *Salmonella* isolates [45] while Ćwiek et al. [36] detected it in 63.6% of isolates. The existence of ESBLs in all samples in our study indicated a shortage of infection barriers between humans and broilers.

Integrons are genetic elements that can recognize and capture movable gene cassettes carrying the AMR genes leading to MDR dissemination and limiting the chemotherapeutic options available for infectious diseases [42]. Regarding integrons genes, our results supported that *int*1 (class 1 integrons) was the most frequent in *Salmonellae* [46]. Additionally, the *int*1 gene was found in all *Salmonella* serovars recovered from poultry and humans [47], while it was found in another study in 82% of isolates [48]. The presence of *int*1 in *Salmonellae* was significantly associated with their AMR and this clearly indicates their important role in the dissemination of AMR genes [42].

The results recorded in this study strongly indicate a correlation between virulence genes and resistance to the most commonly used antimicrobial agents. The relation between virulence and AMR among *Salmonella* serovars occurred because of genetic determinants of AMR- and virulence-associated genes [49]. AMR- and virulence-associated genes may be linked in the same replicon [50]. Finally, the detection of chickens-to-human *Salmonella* strains’ transmission during farming was widely demonstrated and this may occur through the food chain or by direct contact with live broiler chickens during industrial production [25,51].

## 5. Conclusions

ESBLs producing virulent *Salmonella* serovars are a hazard to food safety and public health, creating severe therapeutic problems in the future. The relationship between human and broiler chicken strains strongly indicates that the poultry sector may be considered a very important reservoir for ESBL-producing *Salmonella,* which transmits to humans through the food chain and/or direct contact. However, further studies are badly needed to evaluate the expression of virulence genes using qPCR and compare them with non-virulent serotypes devoid of ESBL genes, which may indicate a variable measurable quantitative analysis of the virulence of the serovars isolated from the chicken farms in order to assess a correlation with the presence and activity of ESBL.

## Figures and Tables

**Table 1 pathogens-11-01196-t001:** Oligonucleotide primers of virulence and resistance genes used in PCR.

Primers	Primer Sequence (5′-3′)		Amplified Product	Reference
*inv*A	F R	GTGAAATTATCGCCACGTTCGGGCAA TCATCGCACCGTCAAAGGAACC	284	[18]
*Omp*A	F R	AGTCGAGCTCATGAAAAAGACAGCTATCGC AGTCAAGCTTTTAAGCCTGCGGCTGAGTTA	1052	[8]
*adr*A	F R	ATGTTCCCAAAAATAATGAA TCATGCCGCCACTTCGGTGC	1113	[19]
*csg*D	F R	TTACCGCCTGAGATTATCGT ATGTTTAATGAAGTCCATAG	651	
*bla* _OXA-1_	F R	ATATCTCTACTGTTGCATCTCC AAACCCTTCAAACCATCC	619	[20]
*bla* _TEM_	F R	ATCAGCAATAAACCAGC CCCCGAAGAACGTTTTC	516	
*bla* _CTX_	F R	ATGTGCAGYACCAGTAARGTKATGGC TGGGTRAARTARGTSACCAGAAYCAGCG	593	[21]
*bla* _MOX_	F R	GCTGCTCAAGGAGCACAGGAT CACATTGACATAGGTGTGGTGC	520	[22]
*int*1	F R	CCTCCCGCACGATGATC TCCACGCATCGTCAGGC	280	[23]
*int*2	F R	TTATTGCTGGGATTAGGC ACGGCTACCCTCTGTTATC	250	
*int*3	F R	AGTGGGTGGCGAATGAGTG TGTTCTTGTATCGGCAGGTG	484	
*qnr*A	F R	ATTTCTCACGCCAGGATTTG GATCGGCAAAGGTTAGGTCA	516	[24]
*qnr*S	F R	ACGACATTCGTCAACTGCAA TAAATTGGCACCCTGTAGGC	417	

**Table 2 pathogens-11-01196-t002:** *Salmonella* serotypes isolated from broiler chicken and human fecal samples.

*Salmonella* Serotype	No. of Isolates	%
Broiler chicken isolated serotypes (*n* = 30)		
*S. enteritidis*	6	20
*S. infantis*	5	16.6
*S. typhimurium*	3	10
*S. kentucky*	3	10
*S. heidelberg*	3	10
*S. hader*	3	10
*S. newport*	3	10
*S. blegdam*	2	6.6
*S. gueuletapee*	1	3.3
*S. maloe*	1	3.3
Human isolated serotypes (*n* = 14)		
*S. enteritidis*	5	35.7
*S. typhimurium*	5	35.7
*S. anatum*	3	21.4
*S. derby*	1	7.1

%: was calculated according to the corresponding No. of isolated serotypes.

**Table 3 pathogens-11-01196-t003:** Antimicrobial susceptibility of *Salmonella* serovars isolated from broiler chickens and humans.

	Broiler Chicken *Salmonella* Isolates (*n* = 30)				Human *Salmonella* Isolates (*n* = 14)			
	Sensitive		Resistant		Resistant		Resistant	
	No.	%	No.	%	No.	%	No.	%
ATM	9	30	21	70	12	85.7	2	14.3
IPM	5	16.6	25	83.3	12	85.7	2	14.3
CTX	6	20	24	80	13	92.8	1	7.2
AM	16	53.3	14	46.6	12	85.7	2	14.3
CAZ	14	46.6	16	53.3	14	100	--	--
CN	21	70	9	30	12	85.7	2	14.3
CRO	8	26.6	22	73.3	12	85.7	2	14.3
CL	22	73.3	8	26.6	13	92.8	1	7.2
AK	12	40	18	60	14	100	--	--
SAM	17	56.6	13	43.3	11	78.5	3	21.4
AMC	14	46.6	16	53.4	13	92.8	1	7.2
SXT	10	33.3	20	66.6	14	100	--	--
CIP	10	33.3	20	66.6	13	92.8	1	7.2

%: was calculated according to the corresponding No. of tested *Salmonella* isolates. AK (Amikacin), AMC (Amoxicillin-clavulanic), AM (Ampicillin), SAM (Ampicillin-Sulbactam), ATM (Aztreonam), CTX (Cefotaxime), CAZ (Ceftazidime), CRO (Ceftriaxone), CL (Cephalexin), CIP (Ciprofloxacin), CN (Gentamicin), IPM (Impenem), SXT (Sulfamethoxazole-trimethoprim).

**Table 4 pathogens-11-01196-t004:** Distribution of virulence and resistance genes in *Salmonella* serovars isolated from broilers and humans.

	*inv*A	*omp*A	*csg*D	*adr*A	*bla* _OXA-1_	*bla* _TEM_	*bla* _CTX_	*bla* _MOX_	*int*1	*int*2	*int*3	*qnr*A	*qnr*S
	MDR *Salmonella* Serovars from Broilers (*n* = 18)												
*S. enteritidis*	+	+	+	+	-	+	-	-	-	-	-	-	-
*S. enteritidis*	+	+	+	+	-	+	-	-	-	-	-	-	-
*S. infantis*	+	+	+	-	-	+	-	-	-	-	-	-	-
*S. infantis*	+	+	+	-	-	+	-	-	-	-	-	-	-
*S. typhimurium*	+	+	+	-	-	+	-	-	-	-	-	-	-
*S. typhimurium*	+	+	+	+	-	+	-	-	-	-	-	-	-
*S. kentucky*	+	+	+	+	-	+	-	-	-	-	-	-	-
*S. kentucky*	+	+	+	+	-	+	-	-	-	-	-	-	-
*S. hedeilberg*	+	+	+	+	-	+	-	-	-	-	-	-	-
*S. hedeilberg*	+	+	+	+	-	+	-	-	-	-	-	-	-
*S. hader*	+	+	+	+	-	+	-	-	+	-	-	-	+
*S. hader*	+	+	+	+	-	+	-	-	+	-	-	-	+
*S. blegdam*	+	+	+	+	-	+	-	-	-	-	-	-	-
*S. blegdam*	+	+	+	+	-	+	-	-	-	-	-	-	-
*S. newport*	+	+	+	+	-	+	-	-	+	-	-	-	-
*S. newport*	+	+	+	+	-	+	+	-	+	-	-	-	-
*S. maloe*	+	+	+	+	-	+	+	-	+	-	-	-	-
*S. gueuletapee*	+	+	+	+	-	+	-	-	-	-	-	-	-
Total no. %	18 100	18 100	18 100	15 83.3	0 0	18 100	2 11.1	0 0	5 27.8	0 0	0 0	0 0	2 11.1
	MDR *Salmonella* Serovars from Human (*n* = 4)												
*S. enteritidis*	+	+	+	+	-	+	-	-	+	-	+	-	-
*S. typhimurium*	+	+	+	+	-	+	-	-	+	-	+	-	-
*S. anatum*	+	+	+	+	-	+	-	-	+	-	+	-	-
*S. derby*	+	+	+	+	-	+	-	-	+	-	+	-	-
Total no. %	4 100	4 100	4 100	4 100	0 0	4 100	0 0	0 0	4 100	0 0	4 100	0 0	0 0

**Table 5 pathogens-11-01196-t005:** Correlation between virulence genes and antimicrobial resistance between *Salmonella* serovars isolated from broilers and Human.

MDR *Salmonella* Serotypes from Broilers			
Tested *Salmonella* Serovars	Virulence Genes Detected	Antimicrobial Resistance Pattern	Resistance Genes Detected
*S. enteritidis*	*inv*A, *adr*A, *csg*D, *omp*A	ATM, IPM, CTX, CAZ, CRO, AK, SXT, CIP	*bla* _TEM_
*S. enteritidis*	*inv*A, *adr*A, *csg*D, *omp*A	ATM, IPM, CTX, AK, SAM, AMC, SXT, CIP	*bla* _TEM_
*S. typhimurium*	*inv*A, *adr*A, *csg*D, *omp*A	ATM, IPM, CTX, AM, CAZ, CN, CRO, SAM, SXT, CIP	*bla* _TEM_
*S. typhimurium*	*inv*A, *csg*D, *omp*A	CTX, CAZ, CRO, AK, SAM, AMC, SXT, CIP	*bla* _TEM_
*S. infantis*	*inv*A, *csg*D, *omp*A	AMC, CAZ, SXT	*bla* _TEM_
*S. infantis*	*inv*A, *csg*D, *omp*A	IMP, AM, SXT	*bla* _TEM_
*S. kentucky*	*inv*A, *adr*A, *csg*D, *omp*A	IMP, CTX, CN, CRO, AK, CIP	*bla* _TEM_
*S. kentucky*	*inv*A, *adr*A, *csg*D, *omp*A	IPM, CTX, CRO, SXT, CIP	*bla* _TEM_
*S. hedeilberg*	*inv*A, *adr*A, *csg*D, *omp*A	ATM, IPM, CTX, CN, CRO, CL, CIP	*bla* _TEM_
*S. hedeilberg*	*inv*A, *adr*A, *csg*D, *omp*A	ATM, IPM, CTX, AM, CAZ, CN, CRO, AK, AMC, SXT, CIP	*bla* _TEM_
*S. hader*	*inv*A, *adr*A, *csg*D, *omp*A	IPM, AM, CIP	*bla*_TEM_, *int*1, *qnr*S
*S. hader*	*inv*A, *adr*A, *csg*D, *omp*A	ATM, IPM, CTX, AM, CRO, CL, AMC, SXT, CIP	*bla*_TEM_, *int*1, *qnr*S
*S. blegdam*	*inv*A, *adr*A, *csg*D, *omp*A	ATM, IMP, CTX, CAZ, CRO, CL, AK, SAM, AMC	*bla* _TEM_
*S. blegdam*	*inv*A, *adr*A, *csg*D, *omp*A	ATM, IMP, CTX, CAZ, CN, CRO, AK, SAM, AMC, SXT, CIP	*bla* _TEM_
*S. newport*	*inv*A, *adr*A, *csg*D, *omp*A	ATM, IPM, CTX, AM, CRO	*bla*_TEM_, *int*1
*S. newport*	*inv*A, *adr*A, *csg*D, *omp*A	ATM, AM, CRO	*bla*_TEM_, *bla*_CTX_, *int*1
*S. maloe*	*inv*A, *adr*A, *csg*D, *omp*A	CAZ, AM, AK	*bla*_TEM_, *bla*_CTX_, *int*1
*S. gueuletapee*	*inv*A, *adr*A, *csg*D, *omp*A	ATM, IPM, CTX, CN, CL, AK, AMC	*bla* _TEM_
MDR *Salmonella* Serotypes from Human			
*S. enteritidis*	*inv*A, *adr*A, *csg*D, *omp*A	ATM, IPM, CN, CRO, CIP	*bla*_TEM_, *int*1, *int*3
*S. typhimurium*	*inv*A, *adr*A, *csg*D, *omp*A	ATM, IPM, CTX, AM, CN, CRO, CL	*bla*_TEM_, *int*1, *int*3
*S. anatum*	*inv*A, *adr*A, *csg*D, *omp*A	CAZ, AM, SAM	*bla*_TEM_, *int*1, *int*3
*S. derby*	*inv*A, *adr*A, *csg*D, *omp*A	CRO, SAM, AMC	*bla*_TEM_, *int*1, *int*3

## Data Availability

The datasets used and/or analyzed during the current study are available from the corresponding author on reasonable request.

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
