# Peer review of "Genetic Correlation of Virulent Salmonella Serovars (Extended Spectrum β-Lactamases) Isolated from Broiler Chickens and Human: A Public Health Concern"

_pathogens, 2022, doi:10.3390/pathogens11101196_

Round 1

Reviewer 1 Report

In this manuscript, the authors report the presence in broiler chickens and humans of Salmonella serovars resistant to several antibiotics commonly used in the poultry industry. Furthermore, they identified several antibiotic resistance genes, including those for extended spectrum beta lactamases, in the Salmonella isolates, overall, from chickens. In general, the manuscript is well written, and the conclusions are supported by data.

Comments.

-The correlation between the presence of virulence and antibiotic-resistance genes is totally expected since the tested virulence genes are conserved in S. enterica. Thus, this issue must be removed from the manuscript.

-Write correctly Salmonella in all text (italics, first letter capital)

-Use commas in large sentences.

-Lines 45-50. Very large sentence.

-Line 50. Correct: Invasion-related gene (invA) gene…

-Line 54. Correct ompA; it shoud be OmpA

Author Response

Reviewer 1:

In this manuscript, the authors report the presence in broiler chickens and humans of Salmonella serovars resistant to several antibiotics commonly used in the poultry industry. Furthermore, they identified several antibiotic resistance genes, including those for extended spectrum beta lactamases, in the Salmonella isolates, overall, from chickens. In general, the manuscript is well written, and the conclusions are supported by data.

  • Thank you so much sir for your opinion. Unfortunately, your comments will add much value to our manuscript.

Comments.

  • The correlation between the presence of virulence and antibiotic-resistance genes is totally expected since the tested virulence genes are conserved in Senterica. Thus, this issue must be removed from the manuscript.

Corrected to the genetic correlation of virulent Salmonella serovars (extended spectrum β-lactamases) isolated from broiler chickens and human.

  • Write correctly Salmonella in all text (italics, first letter capital). Done. Highlighted in yellow color.
  • Use commas in large sentences.Done.
  • Lines 45-50. Very large sentence. Corrected.
  • Line 50. Correct: Invasion-related gene (invA) gene. Corrected.
  • Line 54. Correct ompA; it shoud be OmpA. Corrected.

Reviewer 2 Report

General Comments and corrections

This study is interesting because it aims to detect virulent and ESBL-producing serovars of Salmonella, using 19 isolates from broilers and humans.

However, in Table 5 the authors do not show a correlation or statistical index that proves a link between the ESBL variables and virulence genes described, so it is not possible to speak of a strong correlation as defined in the discussion, which is unsupported.

Furthermore, most of the virulence genes presented in this manuscript are present in all non- typhoidea Salmonella as described in the discussion. Table 5 only shows those bacteria that have resistance genes detected by PCR, so the susceptibility association that is described in point 3.4 is lost.

So, the correlation is forced, there is no way to discriminate between causality or correlation, that is, having ESBL resistance genes does not imply that there is a cause of the presence of virulent genes, perhaps the way in which it is wanted to be evaluated with a study prior to the virulent capacities of the bacteria worked more than the presence or absence of ESBL genes, for example, to evaluate by qPCR the expression of virulence genes and compare them with those that are not virulent and without the presence of ESBL genes, with this one would have a variable measurable quantitative analysis of the virulence of the serovars isolated from the chicken farm in order to assess a correlation with the presence and activity of ESBL.

Minor correction

Line 46 Salmonella path- / Salmonella without italics
Line 47 salomonella pathogenicity / Salmonella without capital letters or italics Line 55 Salmonella antigenic protein / Salmonella without italics
Line 73 human salmonellae / salmonellae without capital letters or italics
Line 120 S. Newport / S. without italics
Table 3 first row / Two Salmonella without italics
Table 4 row % column blaTEM / 11.) error in the decimal
Line 161 / there must be a line break for the title of table 5
Table 5 row 3 S, Enteritidis / there is a comma instead of a period
Line 166 Salmonellosis / capitalized

Line 173 Salmonella / no italics
Line 177 S. Kentucky / S. without italics
Line 199 csgD / without italics
Line 199-200 Salmonella spp. / no italics
Line 201 Salmonella / no italics
Line 205 Salmonella / no italics
Line 209-210 Salmo-nella / no italics
Line 246 salmonellae / without italics or capital letters

Author Response

Reviewer 2:

This study is interesting because it aims to detect virulent and ESBL-producing serovars of Salmonella, using 19 isolates from broilers and humans.

  • Thank you so much sir for your comments. Unfortunately, these comments will add much value to our manuscript.

General Comments and corrections

  • However, in Table 5 the authors do not show a correlation or statistical index that proves a link between the ESBL variables and virulence genes described, so it is not possible to speak of a strong correlation as defined in the discussion, which is unsupported. Furthermore, most of the virulence genes presented in this manuscript are present in all non- typhoidea Salmonella as described in the discussion.
  • We totally agree with you sir. Really we planned to do the next work in the near future studying this correlation in details to support our theory, but we think that these initial data is very important and may have a strong correlation based on the obtained specific cPCR results. The obtained results are supported by previous studies by Halatsi et al., 2006; Ziech et al., 2016; Hawash et al., 2017; Elkenany et al., 2019; Ćwiek, et al., 2020 [35-39].
  • Table 5 only shows those bacteria that have resistance genes detected by PCR, so the susceptibility association that is described in point 3.4 is lost.
  • Ok sir. The word susceptibility is deleted.
  • So, the correlation is forced, there is no way to discriminate between causality or correlation, that is, having ESBL resistance genes does not imply that there is a cause of the presence of virulent genes, perhaps the way in which it is wanted to be evaluated with a study prior to the virulent capacities of the bacteria worked more than the presence or absence of ESBL genes, for example, to evaluate by qPCR the expression of virulence genes and compare them with those that are not virulent and without the presence of ESBL genes, with this one would have a variable measurable quantitative analysis of the virulence of the serovars isolated from the chicken farm in order to assess a correlation with the presence and activity of ESBL.
  • We appreciate your suggestion sir. Your valuable comment added to the conclusion of this study for the near future research.

Line 274-278: However, further studies are badly needed to the evaluate the expression of virulence genes using qPCR and compare them with non-virulent serotypes devoid from ESBL genes may indicate a variable measurable quantitative analysis of the virulence of the serovars isolated from the chicken farms in order to assess a correlation with the presence and activity of ESBL.

Minor correction

Line 46 Salmonella path- / Salmonella without italics

  • Corrected

Line 47 salomonella pathogenicity / Salmonella without capital letters or italics

  • Corrected

Line 55 Salmonella antigenic protein / Salmonella without italics

  • Corrected

Line 73 human salmonellae / salmonellae without capital letters or italics

  • Corrected

Line 120 S. Newport / S. without italics

  • Corrected

Table 3 first row / Two Salmonella without italics

  • Corrected

Table 4 row % column blaTEM / 11.) error in the decimal

  • Corrected

Line 161 / there must be a line break for the title of table 5

  • Corrected

Table 5 row 3 S, Enteritidis / there is a comma instead of a period

  • Corrected

Line 166 Salmonellosis / capitalized

  • Corrected

Line 173 Salmonella / no italics

  • Corrected

Line 177 S. Kentucky / S. without italics

  • Corrected

Line 199 csgD / without italics

  • Corrected

Line 199-200 Salmonella spp. / no italics

  • Corrected

Line 201 Salmonella / no italics

  • Corrected

Line 205 Salmonella / no italics

  • Corrected

Line 209-210 Salmonella / no italics

  • Corrected

Line 246 salmonellae / without italics or capital letters

  • Corrected

Reviewer 3 Report

In the present manuscript, Orabi et al. investigated the genetic correlation between extended spectrum β-lactamases producing and virulent Salmonella serovars isolated from broiler chickens and humans which is an important topic in a one health context. There is only limited information about the situation in Egypt, so this manuscript may add relevant details. However, I have some major comments: i) Authors did not describe the history of human and poultry samples.// ii) The authors did not make any attempt to test if there is any correlation between the presence of virulence genes and disease and performance at least in birds.// iii) Where is the positive control DNA from strains which harbor the genes that could not be amplified from any of the tested strains? //iv) Regarding antibiotic resistance, it would be nice to see the minimal inhibition concentration (MIC). V) Please provide the supplier of all used media, chemicals, reagents, and kits. i.e (Supplier, City, and Country).

Other minor comments: line 3: Salmonella should be italic // line 23: cPCR should be conventional PCR // Line 25: please add „of“ before chicken isolates, // Line 62: Enterobacteriaceae should be italic, Line 65: I suggest to add this recent reference ( https://doi.org/10.51585/gjvr.2021.2.0011)  // Line 79: Please provide more data about the sampling „for poultry species, age, healthy condition“ for human: hospitalized febrile patients? // Line 80: Please provide the supplier, City, and Country for all used media/reagents/ and kits? // Line 91: Which antisera was used? Please provide also the supplier.// Table 3: Salmonella should be italic.// Line 172: „that that reported….“  Should be „than that reported….)// Line 96: Why „Tetracyclines“not included in this study? To my knowledge, it is still widely used in Egypt.// Line 177: Please add this sentence: „In 2018 The most prevalent serotypes were S. Enteritidis, S. Typhimurium, S. Pullorum, S. Vejle, S. Amsterdam, S. Infantis, S. Petersburg, and S. Atakpame (Shehata et al. 2018; J. http://dx.doi.org/10.29261/pakvetj/2018.xxx )“.// Line 189: Please add also these recent findings: „Recently in Nigeria, 75% of Salmonella isolated from hospitalized febrile patients harbored the invA gene, (https://doi.org/10.51585/gjm.2021.3.0008).

Author Response

Reviewer 3:

In the present manuscript, Orabi et al. investigated the genetic correlation between extended spectrum β-lactamases producing and virulent Salmonella serovars isolated from broiler chickens and humans which is an important topic in a one health context. There is only limited information about the situation in Egypt, so this manuscript may add relevant details.

  • Thank you so much sir for your kind words and comments. Unfortunately, these comments will add much value to our manuscript

Some major comments:

  1. Authors did not describe the history of human and poultry samples.
  • Added to samples, lines 80-91: A total of 360 fecal samples (300 from broiler chicken and 60 from human) were collected in plastic screw-top tubes containing sterile 10 ml buffered peptone water (BPW) (Oxoid LTD, Basing Stoke, UK) under complete hygienic conditions and transported in ice box to the laboratory of faculty of veterinary medicine, Beni-Suef University, for bacteriological examination. Regarding the history of chicken samples, they were collected from 30 broiler chicken flocks resembling 5 Egyptian governorates during 2019-2020 with a total no. of 300,000 aged 1 to 14 days and were suffered from unabsorbed yoik sac, enteritis, diarrhea, arthritis and conjunctivitis. For human samples, all were collected from 60 patients aged 30-40 years during 2020 suffering from enteritis and hepatic problems.

  1. The authors did not make any attempt to test if there is any correlation between the presence of virulence genes and disease and performance at least in birds.
  • We totally agree with you sir. Really, this will be the next work to apply further in-vivo studies in birds to confirm the correlation between the presence of virulence genes and disease production and performance effect.
  • Where is the positive control DNA from strains which harbor the genes that could not be amplified from any of the tested strains?
  • During the PCR step: local isolates previously identified and preserved in the Reference laboratory for veterinary quality control on poultry production, Animal health research institute its DNA used as standard control positive in the present study.
  1. Regarding antibiotic resistance, it would be nice to see the minimal inhibition concentration (MIC).
  • The present aimed to just monitor the genetic correlation between the virulence genes and ESBL genes in the most common isolates either from broilers or human and we promise to complete further study included the MIC  to complete the epidemiological picture of this zoonotic pathogen: Salmonella
  1. Please provide the supplier of all used media, chemicals, reagents, and kits. i.e (Supplier, City, and Country).
  • All are provided.

Other minor comments:

  • Line 3: Salmonella should be italic.
  • Done all over the manuscript.
  • Line 23: cPCR should be conventional PCR.
  • Corrected
  • Line 25: please add „of“ before chicken isolates,
  • Added
  • Line 62: Enterobacteriaceae should be italic.
  • Corrected
  • Line 65: I suggest to add this recent reference (https://doi.org/10.51585/gjvr.2021.2.0011)
  • Added
  • Line 79: Please provide more data about the sampling „for poultry species, age, healthy condition“ for human: hospitalized febrile patients?
  • The history of the samples added
  • Line 80: Please provide the supplier, City, and Country for all used media/reagents/ and kits?
  • All are provided.
  • Line 91: Which antisera was used? Please provide also the supplier.
  • Standard “O" and "H" antisera (Difco, USA). Line 94
  • Table 3: Salmonella should be italic.
  • Done all over the manuscript.
  • Line 172: „that that reported….“  Should be „than that reported….)
  • Corrected
  • Line 96: Why„ Tetracyclines“ not included in this study? To my knowledge, it is still widely used in Egypt.
  • Thank you for your notice sir. Really, the widely misuse of Tetracyclines is the main cause of its exclusion as in most of our research targeting the sensitivity testing of Salmonella or coli isolates, Tetracyclines tested negative.
  • Line 177: Please add this sentence: „In 2018 The most prevalent serotypes were S. Enteritidis, S. Typhimurium, S. Pullorum, S. Vejle, S. Amsterdam, S. Infantis, S. Petersburg, and S. Atakpame (Shehata et al. 2018; J. http://dx.doi.org/10.29261/pakvetj/2018.xxx )“.
  • Lines 185-187.
  • Line 189: Please add also these recent findings: „Recently in Nigeria, 75% of Salmonellaisolated from hospitalized febrile patients harbored the invA gene, (https://doi.org/10.51585/gjm.2021.3.0008).
  • Lines 199-201.

Round 2

Reviewer 2 Report

the authors have responded to all the suggestions of the evaluation 

Reviewer 3 Report

Many thanks to the authors, all comments have been addressed. Accept in the present form.